# Splitting & Integrating: Out-of-Distribution Detection via Adversarial Gradient Attribution

**Jiayu Zhang** [* 1]  **Xinyi Wang** [* 2]  **Zhibo Jin** [3]  **Zhiyu Zhu** [3]  **Jianlong Zhou** [3]  **Fang Chen** [3]  **Huaming Chen** [4]

## Abstract

Out-of-distribution (OOD) detection is essential for enhancing the robustness and security of deep learning models in unknown and dynamic data environments. Gradient-based OOD detection methods, such as GAIA, analyse the explanation pattern representations of in-distribution (ID) and OOD samples by examining the sensitivity of model outputs w.r.t. model inputs, resulting in superior performance compared to traditional OOD detection methods. However, we argue that the non-zero gradient behaviors of OOD samples do not exhibit significant distinguishability, especially when ID samples are perturbed by random perturbations in high-dimensional spaces, which negatively impacts the accuracy of OOD detection. In this paper, we propose a novel OOD detection method called **S & I** based on layer **S**plitting and gradient **I**ntegration via Adversarial Gradient Attribution. Specifically, our approach involves splitting the model's intermediate layers and iteratively updating adversarial examples layer-by-layer. We then integrate the attribution gradients from each intermediate layer along the attribution path from adversarial examples to the actual input, yielding true explanation pattern representations for both ID and OOD samples. Experiments demonstrate that our S & I algorithm achieves state-of-the-art results, with the average FPR95 of 29.05% (ResNet34)/38.61% (WRN40) and 37.31% (BiT-S) on the CIFAR100 and ImageNet benchmarks, respectively. Our code is available at: https://github.com/LMBTough/S-I

[*]Equal contribution [1]Suzhou University of Technology, Suzhou, China [2]Faculty of Computer Science & Information Technology, University of Malaya, Kuala Lumpur, Malaysia [3]Data Science Institute, University of Technology Sydney, Sydney, Australia [4]School of Electrical & Computer Engineering, University of Sydney, Sydney, Australia. Correspondence to: Huaming Chen <huaming.chen@sydney.edu.au>.

*Proceedings of the 42^{nd} International Conference on Machine Learning*, Vancouver, Canada. PMLR 267, 2025. Copyright 2025 by the author(s).

## 1. Introduction

Deep neural networks have achieved remarkable success in a variety of domains, including autonomous driving (Chen et al., 2021) and medical diagnosis (Yadav & Jadhav, 2019). However, their performance and reliability are strongly influenced by the assumption that the test data originates from the same distribution as the training data. In practical applications, this assumption is frequently violated, as models often face inputs that deviate significantly from the in-distribution (ID) training data. Such inputs, known as out-of-distribution (OOD) samples, present a major challenge for deep neural networks, which can produce overconfident yet incorrect predictions.

Therefore, performing OOD detection is essential for ensuring the safe and reliable deployment of deep neural networks in real-world applications. Currently post-hoc OOD detection methods can be mainly divided into three categories: output-based methods (Hsu et al., 2020; Liu et al., 2020; Hendrycks & Gimpel, 2016; Liang et al., 2017), feature representation-based methods (Sun et al., 2021; Sastry & Oore, 2020; Song et al., 2022) and gradient-based methods (Huang et al., 2021; Lee & AlRegib, 2020; Igoe et al., 2022; Chen et al., 2023). Among them, output-based methods rely on the confidence score of the model output to determine whether the input sample belongs to the training data distribution, while feature representation-based methods detect OOD samples by analyzing the feature vectors of the intermediate layers of the neural network. However, compared with gradient-based methods that identify OOD samples by calculating the gradient information of input samples w.r.t. model parameters (or a certain layer output), they are easily deceived by some OOD samples with high output similarity or easily affected by the quality of feature representation. Therefore, in this paper we focus on gradient-based methods.

As one of the mainstream gradient-based methods, the GAIA (Chen et al., 2023) algorithm investigates the explanation pattern representations of ID and OOD samples from the sensitivity of model outputs w.r.t model inputs, i.e., the attribution gradients (Simonyan, 2013). Specifically, by back-propagating the attribution gradient $\frac{\partial f(x;\theta)}{\partial x}$ of the model output $f(x;\theta)$ w.r.t. the input sample $x$ on each inter-

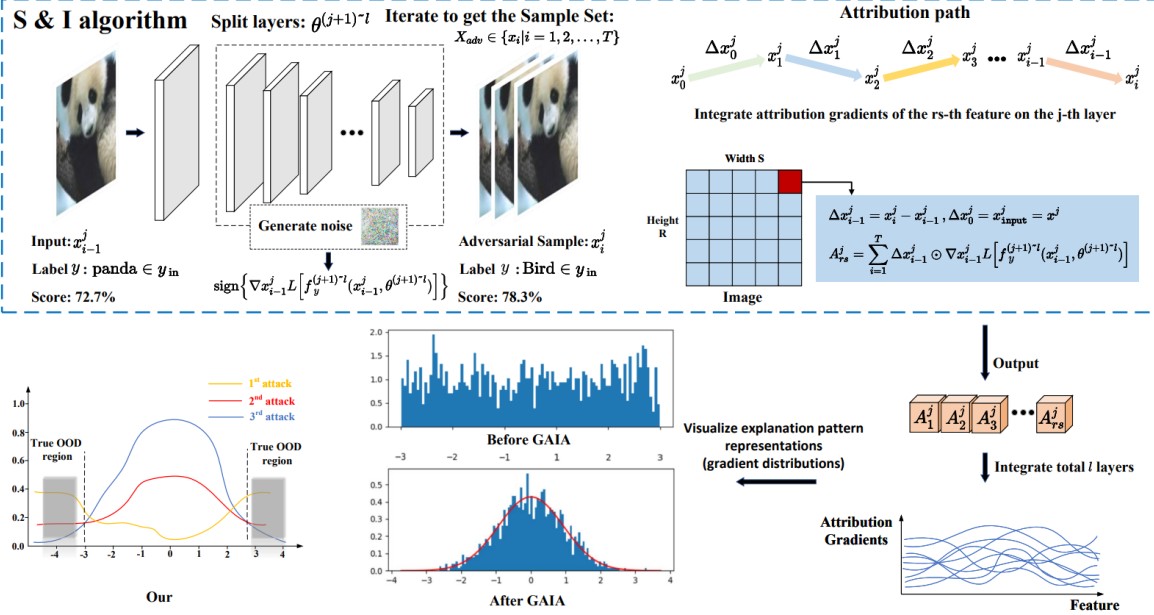

*Figure 1.* Algorithm flowchart. The gradient distribution (Y-axis frequency) of OOD samples investigated by GAIA tends to exhibit non-zero values (X-axis attribution gradient). We argue that the abnormal gradients induced by perturbations in input data cause feature components that should be predicted as ID to be incorrectly classified as OOD, resulting in irregular gradient distributions. By performing multiple adversarial attacks to analyze the feature distribution shifts from ID adversarial examples to OOD input samples, we can progressively identify high-confidence non-zero gradients and obtain true explanation pattern representations denoted by the shaded regions.

mediate layer, GAIA considers input samples with a large number of non-zero attribution gradients as OOD samples. As shown in **Fig. 2**, we find that for OOD samples, the attribution map often does not focus on certain key features and shows a scattered pattern, which means that the model has no clear understanding of OOD samples. Therefore, we argue that this phenomenon indicates that the model may have higher sensitivity (i.e., larger gradient value) to any feature under an unseen distribution, and even some irrelevant details will get high gradient values. This characteristic makes the non-zero gradient behavior of OOD samples not significantly differentiating, especially when ID samples are subject to random perturbations in high-dimensional space. At this time, the gradient fluctuation caused by small input changes of the model will make it difficult for the gradient sensitivity to stably reflect the actual relationship between the model output and the input, affecting the distinction between the explanation pattern representations of ID and OOD samples.

In this paper, to address this shortcoming, for the first time, we investigate the explanation pattern representation of ID and OOD samples from the perspective of adversarial attacks (Kurakin et al., 2018). Specifically, we introduce adversarial examples to artificially add perturbations to input samples. Then, we use adversarial examples as baselines and gradually **integrating** the adversarial gradi-

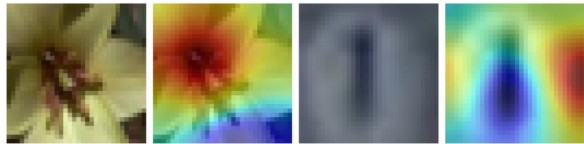

*Figure 2.* Attribution visualization. The left two images (label 'tulip') represent the ID input sample and its attribution map, while the right two images (label '0') represent the OOD input sample and its attribution map.

ent $\frac{\partial L(f(x_i;\theta))}{\partial x_i}$ of the loss function over the model output $L(f(x_i;\theta))$ w.r.t the $i$-th iteration adversarial example $x_i$ along the attribution path from the baseline to the actual input, thereby smoothing the volatility of the attribution gradient and reflecting the true explanation pattern representation.

Besides, it is worth emphasizing that traditional gradient-based methods such as GAIA assume that the influence of each intermediate layer of the model on the input features is uniform and linearly cumulative. In fact, the sensitivity of intermediate features in different layers to the input may be highly heterogeneous, with early layers focusing on low-level edge or texture information and later layers focusing on high-level semantic features. In deeper neural networks, this may introduce unstable gradient explosions

or cumulative errors in inter-layer features, reducing the representation accuracy of the explained pattern. To address this problem, we introduce the concept of layer **splitting** for the first time. Assuming that the neural network has a total of $l$ intermediate layers, we split the current $j$-th intermediate layer from the subsequent $(j + 1 \sim l)$-th intermediate layer while updating the adversarial example layer by layer. Based on these insights, we propose a novel OOD detection method called **S & I** based on layer **S**plitting and gradient **I**ntegration via Adversarial Gradient Attribution. Comprehensive experiments on both CIFAR100 and large-scale ImageNet-1K benchmarks validate the effectiveness of our S & I algorithm. **Fig. 1** shows the algorithm flowchart.

Our key contributions are summarized as follows:

- Given the observation that the attribution gradients of OOD samples are not significantly distinguishable, in order to reduce the abnormal gradient fluctuations caused by random perturbations in ID samples in high-dimensional space, we first introduce adversarial examples to artificially add perturbations to the input samples for OOD detection, thereby reflecting ture explanation pattern representations.

- We, for the first time, propose the concept of layer splitting and adversarial attribution gradient integration for OOD detection. By decomposing intermediate layers and iteratively updating adversarial examples layer-by-layer, we integrate the attribution gradients of each iteration along the attribution path from adversarial examples to the actual input sample. We also give the theoretical proof of our algorithm in our paper.

- Experiments demonstrate that our S & I algorithm achieves SOTA results, with the average FPR95 of 29.05% (38.61%) and 37.31% on the CIFAR100 and ImageNet benchmarks, respectively. We have also open-sourced the relevant code.

## 2. Preliminaries

### 2.1. Problem definition

Given a deep neural network $f$ with parameters $\theta$, for a supervised task, the output of the network for the input sample space $X$ can be expressed as $f(X; \theta; Y)$. Here $Y$ represents the label space, and in the following we omit $Y$ for convenience. The goal of out-of-distribution (OOD) detection is to identify input data that comes from a distribution different from the training data. Let $x_{in} \in X$ represents the in-distribution (ID) samples, and $x_{out} \in X$ represents the OOD samples. Typically, there is no intersection between the label sets $y_{in} \in Y$ and $y_{out} \in Y$ for ID and OOD samples. Taking an image classification task as an example, since the model $f$ has never seen OOD data $x_{out}$ during

training, it tends to produce overconfident predictions for such inputs (Nguyen et al., 2015; Hein et al., 2019). Based on this characteristic, OOD detection can be formulated as a binary classification problem as follows (Zhang et al., 2023):

$$Binary\ Classifier = \begin{cases} OOD & , if \quad \Omega(X) \geq \xi \\ ID & , if \quad \Omega(X) < \xi \end{cases}$$
$$(1)$$

Where $\xi$ represents the threshold for distinguishing OOD and ID samples, and $\Omega(X)$ is the confidence score function for the binary classification. We consider input samples $x$ with confidence scores greater than or equal to $\xi$ as OOD samples $x_{out}$. Here Eq. 1 provides a mathematical understanding of OOD detection and does not affect the subsequent introduction of our algorithm.

### 2.2. From gradient-based attribution to adversarial attack

In general, for an image classification interpretation task, the objective of gradient-based attribution is to determine an attribution value $A_{rs} \in \mathbb{R}^{R \times S \times K}$ that reflects the importance of each feature component $\underset{(rs)}{x}$ within the input sample $x \in \mathbb{R}^{R \times S \times K}$ w.r.t. the model output $f(x; \theta)$. Here $\underset{(rs)}{x}$ is a pixel-level feature, $S$ and $R$ represent the width and height of the $k$-th channel input sample. $f(x; \theta)$ typically represents the predicted labels of the image, expressed as confidence scores for each class.

One approach to understanding how a model makes decisions is to pinpoint the minimal feature changes that either weaken or strengthen its current prediction. This requires that the feature modifications remain limited, so as not to distort the semantic content of the original sample. Consequently, the challenge of interpretation can be reformulated as identifying the most influential features that affect the model's decision, while ensuring the changes remain within certain constraints.

**Attribution gradients calculation** Currently, commonly employed gradient-based attribution algorithms, such as Integrated Gradients (IG) (Sundararajan et al., 2017) and Boundary-based Integrated Gradients (BIG) (Wang et al., 2021), utilize gradient information $\frac{\partial f(x;\theta)}{\partial x}$ to represent local changes for calculating importance scores. If we denote the importance of each feature component in the input sample calculated by IG as $A_{rs}^{IG}$, then the integration process of IG can be expressed as Eq. 2:

$$A_{rs}^{IG}(x) = \left( \underset{(rs)}{x} - \underset{(rs)}{x'} \right) \times \sum_{i=1}^{T} \frac{\partial f \left( x' + \frac{i}{T} \times (x - x') \right)}{\partial \underset{(rs)}{x}} \times \frac{1}{T} \tag{2}$$

where $rs = 1, 2, ..., RS$ represents the $rs$-th feature component in the input sample $x$. The gradient of the model output w.r.t. the $rs$-th feature component is denoted by $\frac{\partial f\left(x' + \frac{i}{T} \times (x - x')\right)}{\partial \underset{(rs)}{x}}$. In this context, $x'$ denotes the baseline sample, typically represented by a black image or a zero embedding vector in image or text models. From Eq. 2, we can see IG divides the integration path $(x - x')$ into $T$ iterations with equal step sizes to compute $A_{rs}^{IG}(x)$. In GAIA (Chen et al., 2023), the authors argue that the attribution gradients $g = \frac{\partial f\left(x' + \frac{i}{T} \times (x - x')\right)}{\partial \underset{(rs)}{x}}$ related to the input samples are the key gradients for OOD detection. Moreover, input samples $x$ exhibiting non-zero attribution gradients across most feature components $\underset{(rs)}{x}$ are highly likely to be OOD samples.

**Accuracy loss of attribution gradients** However, both attribution algorithm IG or attribution-based OOD detection algorithm GAIA set the baseline sample $x'$ as a black image, i.e., $x' = 0$. It is worth noting that for tasks of varying scales, the selection of baseline points is complex and often ad-hoc. Additionally, using black images as baselines can make it difficult to preserve the original semantic information. In this regard, adversarial attacks (Kurakin et al., 2018)—capable of altering model decisions with minimal perturbations—can generate adversarial examples that are highly similar to the original images, relying solely on input samples and the model. Therefore, employing adversarial examples as baselines for attribution retains semantic information and eliminates the need for a specific baseline selection method. We believe that using adversarial examples with semantics similar to the original sample as the baseline (Pan et al., 2021; Zhu et al., 2024b;a) can improve the accuracy of attribution gradient calculations, a concept that has already been demonstrated in several SOTA attribution algorithms. The accuracy of attribution gradients is crucial for attribution-based OOD detection, as it significantly influences the distribution of the attribution gradients.

### 2.3. Definition of adversarial attacks

Given a deep neural network $f$ and an original input sample $x \in \mathbb{R}^{R \times S \times K}$, for a standard image classification task, where the true label corresponding to $x$ is $t \in y_{in}$, the objective of adversarial attacks is to generate an adversarial example $x_{adv}$ by adding perturbations to $x$. These perturbations are designed to mislead the model into making incorrect

predictions while maintaining the semantic similarity to the original input. In this scenario, the label of the adversarial example is manipulated to be $t'$. It is important to note that, according to the characteristic of adversarial attacks, the label $t'$ is manipulated by the model during training, and therefore, $t'$ still belongs to the ID label set $y_{in}$. Generally, $T$ iterations are required to obtain the optimal adversarial example. The attack process can be described as follows:

$$x_i = x_{i-1} + \eta \cdot sign\left(\nabla_{x_{i-1}} L(f(x_{i-1}); \theta)\right)$$
$$s.t. \quad f(x_i; \theta) = t' \neq t \tag{3}$$

where $\eta$ denotes the learning rate, $i = 1, 2, \ldots, T$, $x_0 = x$, and $x_{adv} = x_T$. The $sign(\cdot)$ function indicates the direction of the update for the adversarial example. To ensure that the perturbations added do not alter the semantic information of the original sample, we constrain the magnitude of these perturbations as follows:

$$\|x_{adv} - x\|_2 \leq \epsilon \tag{4}$$

where $\|\cdot\|_2$ represents the $L_2$ norm and $\epsilon$ denotes the maximum allowable perturbation. It is clear that the iteration of adversarial samples can be interpreted as a gradient ascent process that maximizes the loss function associated with the original label (thereby misleading the model's predictions) while simultaneously minimizing the perturbations applied to the input sample, in accordance with the requirements of the interpretation challenge. In the next section, we will introduce how we incorporate adversarial attacks into attribution to explore the distributional characteristics of ID and OOD samples.

## 3. Layer splitting and adversarial attribution gradient integration for OOD detection

### 3.1. Zero importance verification under the adversarial attack

In this subsection, we first give a mathematical proof of zero importance verification under the adversarial attack. Our goal is to proof that, when adversarial examples are used as the baseline, the attribution gradients of each feature component $\underset{(rs)}{x}$ still tend to be zero for ID samples, indicating zero importance. In the GAIA (Chen et al., 2023) scenario, we can express the model output $f(x; \theta)$ w.r.t the true label $t$ using a higher-order Taylor expansion under the zero baseline (black image):

$$f(x;\theta) = f(0;\theta) + \sum_{p=1}^{P} \sum_{rs=1}^{RS} \frac{1}{p!} \frac{\partial^p f(x;\theta)}{\partial \underset{(rs)}{x}^p} \underset{(rs)}{x}^p$$
$$+ \frac{1}{2!} \frac{\partial^2 f(x;\theta)}{\partial \underset{(1)}{x} \partial \underset{(2)}{x}} \underset{(1)(2)}{x x} + ... + R_p(x) \quad (5)$$

where $\frac{\partial^p f(x;\theta)}{\partial \underset{(rs)}{x}^p}$ represents the $p$-th order derivative of output $f(x;\theta)$ w.r.t the feature component $\underset{(rs)}{x}$. $\frac{\partial^2 f(x;\theta)}{\partial \underset{(1)}{x} \partial \underset{(2)}{x}}$ represents the second-order mixed partial derivative of $f(x;\theta)$. $R_p(x)$ is the remainder after Taylor expansion. Then we can get the following label output change, i.e., the absolute value of the attribution for the input sample $x$:

$$|A(x)| = |f(x;\theta) - f(0;\theta)|$$
$$= \left| \sum_{p=1}^{P} \sum_{rs=1}^{RS} \frac{1}{p!} \frac{\partial^p f(x;\theta)}{\partial \underset{(rs)}{x}^p} \underset{(rs)}{x}^p \right|$$
$$+ \left| \frac{1}{2!} \frac{\partial^2 f(x;\theta)}{\partial \underset{(1)}{x} \partial \underset{(2)}{x}} \underset{(1)(2)}{x x} + ... + R_p(x) \right| \quad (6)$$

According to the description of the sensitivity axiom in GAIA and IG (Sundararajan et al., 2017), we can get the following theorem:

**Theorem 1:** An attribution method adheres to the *Sensitivity Axiom* if, for any input and baseline that differ in a single feature and produce different predictions, the feature with the difference must be assigned a non-zero attribution.

Since GAIA demonstrates that OOD samples typically exhibit overconfident predictions (Chen et al., 2023), we can assert that the label output change for OOD samples $|f(x_{out};\theta) - f(0;\theta)|$ is, to some extent, greater than that $|f(x_{in};\theta) - f(0;\theta)|$ for ID samples. Then we can get $|A(x_{in})| < |A(x_{out})|$ in common cases. This is intuitive because the feature components of ID samples typically match the distribution of the training data, resulting in a smaller contribution to the predictions and relatively lower attribution values. According to *Theorem 1*, *the attribution for features that do not influence the model predictions is zero, indicating zero importance*. Therefore, the smaller attribution of ID samples $|A(x_{in})| = \sum_{rs=1}^{RS} |A_{rs}(x_{in})|$ imply that the gradient polynomials associated with the feature components $\underset{(rs)}{x_{in}}$ in the higher-order Taylor expansion have a higher occurrence of zero gradients.

**Proposition 1:** For a feature component $\underset{(rs)}{x} \in x$ that is to be attributed, if $\frac{\partial f(x;\theta)}{\partial \underset{(rs)}{x}}$ is zero throughout the entire attribution process, then $|A_{rs}(x)|=0$. In this case, the input

sample $x$ with a higher prevalence of zero-valued $\frac{\partial f(x;\theta)}{\partial \underset{(rs)}{x}}$ yield smaller attribution $|A(x)| = \sum_{rs=1}^{RS} |A_{rs}(x)|$, indicating an ID sample $x_{in}$.

**Proof 1:** It is known from advanced calculus that if $\frac{\partial f(x;\theta)}{\partial \underset{(rs)}{x}} = 0$, then its $p$-th partial derivative $\frac{\partial^p f(x;\theta)}{\partial \underset{(rs)}{x}^p} = 0$. Consequently, due to the chain rule of gradients, its $p$-th mixed partial derivative $\frac{\partial^p f(x;\theta)}{\partial \underset{(1)}{x} \partial \underset{(2)}{x} ... \partial \underset{(p)}{x}} = 0$. From Eq. 6, $|A_{rs}(x)| = 0$ always holds.

$$|A(x)| = |f(x;\theta) - f(x_{adv};\theta)|$$
$$= \left| \sum_{p=1}^{P} \sum_{rs=1}^{RS} \frac{1}{p!} \frac{\partial^p f(x;\theta)}{\partial \underset{(rs)}{x}^p} (\underset{(rs)}{x_{adv}} - \underset{(rs)}{x})^p \right|$$
$$+ \left| \frac{1}{2!} \frac{\partial^2 f(x;\theta)}{\partial (\underset{(1)}{x_{adv}} - \underset{(1)}{x}) \partial (\underset{(2)}{x_{adv}} - \underset{(2)}{x})} \right. \quad (7)$$
$$\left. \cdot (\underset{(1)}{x_{adv}} - \underset{(1)}{x})(\underset{(2)}{x_{adv}} - \underset{(2)}{x}) \right|$$
$$+ \cdots + |R_p(x_{adv} - x)|$$

In the adversarial attack scenario, instead of using $f(0;\theta)$, we use $f(x_{adv};\theta)$ as the baseline. At this time, Eq. 6 is transformed into Eq. 7.

**Proposition 2:** When the baseline sample is an adversarial sample, if the gradient $\frac{\partial f(x;\theta)}{\partial \underset{(rs)}{x}}$ satisfies the conditions in *Proposition 1*, then the attribution gradients of each feature component $\underset{(rs)}{x}$ still tend to be zero for ID samples.

**Proof 2:** After adversarial attacks, the label $t'$ of the adversarial sample still belongs to the ID label set $y_{in}$. Additionally, adversarial samples possess the characteristic that require iterative training within the neural network. Therefore, adversarial samples can be regarded as ID samples in our opinion. According to **Theorem 1**, GAIA demonstrates that OOD samples typically exhibit overconfident predictions, since the input sample $x \in x_{in}$, then neither $f(x;\theta)$ nor $f(x_{adv};\theta)$ exhibits overly confidence in this case. We can get a low-level $|A(x)|$, which means that the input sample $x$ has a higher prevalence of zero-valued $\frac{\partial f(x;\theta)}{\partial \underset{(rs)}{x}}$. When the input sample $x \in x_{out}$, then $f(x_{adv};\theta)$ will exhibit overly confidence. In this case, we can demonstrate that $|f(x_{out};\theta) - f(x_{adv};\theta)| > |f(x_{in};\theta) - f(x_{adv};\theta)|$, indicating a higher prevalence of non-zero gradients for OOD samples.

## 3.2. S & I algorithm

### 3.2.1. ADVERSARIAL ATTRIBUTION GRADIENT INTEGRATION

From Sec. 3.1, it can be concluded that the key to OOD detection lies in obtaining the distribution of attribution gradients. For the input sample $x$, we perform Eq. 3 to update adversarial examples. To integrate the attribution gradients we need, we apply the first-order Taylor approximation to expand the loss function and incorporate the gradient information along the attribution path from $x_0$ to $x_T$:

$$L\left(f\left(x_i\right)\right) = L\left(f\left(x_{i-1}\right)\right) + \frac{\partial L\left(f\left(x_{i-1}\right)\right)}{\partial x_{i-1}}\left(x_i - x_{i-1}\right) + o$$

$$\sum_{i=1}^{T} L\left(f\left(x_i\right)\right) = \sum_{i=0}^{T-1} L\left(f\left(x_i\right)\right) + \sum_{i=0}^{T-1} \frac{\partial L\left(f\left(x_i\right)\right)}{\partial x_i}\left(x_{i+1} - x_i\right)$$

$$A = L\left(f\left(x_T\right)\right) - L\left(f\left(x_0\right)\right) = \sum_{i=0}^{T-1} \frac{\partial L\left(f\left(x_i\right)\right)}{\partial x_i}\left(x_{i+1} - x_i\right)$$

$$= \sum_{i=0}^{T-1} \triangle x_i \odot g(x_i) = \sum_{i=1}^{T} \triangle x_{i-1} \odot g(x_{i-1})$$

$$(8)$$

Here $o$ and $\theta$ is omitted for convenience. And $\triangle x_{i-1} = x_i - x_{i-1}$, $g(x_{i-1}) = \frac{\partial L(f(x))}{\partial x_{i-1}}$. It is obvious that Eq. 8 satisfies *Theorem 1*. However, there is a problem with Eq. 8. Since the neural network has $l$ intermediate layers, we cannot use the union parameters $\theta$ of the neural network when performing gradient ascent on the $j$-th layer. In fact, we use the parameters $\theta^{(j+1)\sim l}$ of the $(j+1) \sim l$-th layers to update the adversarial examples. Unlike GAIA, which assumes that each intermediate layer of the model has a uniform impact on the feature map, our purpose is to distinguish the sensitivity of intermediate feature maps on different layers to the model input. Therefore, we first introduce the concept of layer splitting to deeply investigate the distribution of attribution gradients.

### 3.2.2. LAYER SPLITTING

Specifically, assuming that the dimension of the sample space is $\mathbb{R}^{R \times S \times K}$, we will use the following formula to update the adversarial example $x_i^{jk}$ with predicted label $y$ on the $k$-th channel, $j$-th layer:

$$x_i^{jk} = x_{i-1}^{jk} + \eta \cdot sign\left(\frac{\partial L\left(f_y^{(j+1)\sim l}\left(x_{i-1}^{jk};\theta^{(j+1)\sim l}\right)\right)}{\partial x_{i-1}^{jk}}\right)$$

$$(9)$$

where $x_0^{jk} = x^{jk}$. And we can get $\triangle x_{i-1}^{jk} = x_i^{jk} - x_{i-1}^{jk}$, $g(x_{i-1}^{jk}) = \frac{\partial L\left(f_y^{(j+1)\sim l}\left(x_{i-1}^{jk};\theta^{(j+1)\sim l}\right)\right)}{\partial x_{i-1}^{jk}}$. To compute attribution of the $rs$-th feature component on $x^{jk}$, we then trans-

form Eq. 8 into:

$$A_{rs}^{jk} = \sum_{i=1}^{T} \triangle x_{i-1}^{jk}_{(rs)} \odot g(x_{i-1}^{jk})_{(rs)}$$

$$(10)$$

From **Proposition 2**, it can be deduced that if the attribution gradient $g(x_{i-1}^{jk})_{(rs)}$ of the feature component $x_{i-1}^{jk}_{(rs)}$ on the $j$-th layer and the $k$-th channel tends to be non-zero, then the feature component tends to be OOD. Therefore, we need to compute the non-zero density of input sample $x^{jk}$ to obtain the non-zero expectation. Here, following the conditions set by GAIA-Z, when the label space $Y$ is relatively small, such as in CIFAR100 (Krizhevsky et al., 2009), we can derive the expectation:

$$E\left[\epsilon|x^{jk}\right] = \frac{1}{R \times S \times T}\left|\left\{x_{i-1}^{jk}_{(rs)}|g(x_{i-1}^{jk})_{(rs)} \neq 0\right\}\right|$$

$$(11)$$

When the labe space $Y$ is relatively large, such as ImageNet (Deng et al., 2009), it is time-consuming to calculate the non-zero density for each label in the dataset. Following the conditions set by GAIA-A, assuming that the network feature extraction function is $\Psi(\cdot)$, we can get the last $l$-th layer input $x_{i-1}^l = \Psi(x_{i-1};\theta)$. Considering the gradient matrix on the $l$-th layer, $k$-th channel input sample $x_{i-1}^{lk}$ and the $j$-th layer, $k$-th channel input sample $x_{i-1}^{jk}$, we get:

$$\triangledown x_{i-1}^{jk} = \frac{\partial x_{i-1}^{lk}}{\partial x_{i-1}^{jk}}$$

$$\triangledown x_{i-1}^{lk} = \frac{\partial\left(0.9 * \sum_{y_m \in Y}\left(\log \text{softmax}\left(f_{y_m}^l\left(x_{i-1}^{lk};\theta^l\right)\right)\right)\right)}{\partial x_{i-1}^{lk}}$$

$$(12)$$

where $Y = \{y_m|y_m \in Y\}$. It is worth noting that unlike GAIA, we only take the top-$N\%$ outputs (here top-90%) when integrating the outputs of each label to remove the influence of redundant channels in the last layer. Then we can get the expectation w.r.t. $x^{jk}$:

$$E\left[\epsilon|x^{jk}\right] = \frac{\left|\frac{1}{R \times S \times T}\sum_{i=1}^{T}\sum_{G_{i-1}^{jk} \in \triangledown x_{i-1}^{jk}}\left(G_{i-1}^{jk}\right)\right|}{\left|\frac{1}{R^l \times S^l \times T}\sum_{i=1}^{T}\sum_{G_{i-1}^{lk} \in \triangledown x_{i-1}^{lk}}\left(G_{i-1}^{lk}\right)\right|^{\frac{1}{2}}}$$

$$(13)$$

where $R^l$ and $S^l$ represent the height and width of the last $l$-th layer input sample, respectively. $G$ represents a gradient component in the gradient matrix. Finally, we can get the overall OOD score:

*Table 1.* Experimental result on CIFAR100 benchmark. Here backbone models are ResNet34 and WRN40. The lower the FPR95, the better the performance, with AUROC behaves inversely. All values are percentages and the best value is bolded.

| Dataset/Model | Methods | SVHN | | LSUN | | TinyImageNet | | Places | | Textures | | AVG | |
|---|---|---|---|---|---|---|---|---|---|---|---|---|---|
| | | FPR95↓ | AUROC↑ | FPR95↓ | AUROC↑ | FPR95↓ | AUROC↑ | FPR95↓ | AUROC↑ | FPR95↓ | AUROC↑ | FPR95↓ | AUROC↑ |
| CIFAR100 /ResNet34 | MSP | 85.69 | 74.8 | 83.87 | 73.7 | 78.05 | 77.11 | 86.4 | 72.65 | 82.09 | 74.79 | 83.22 | 74.61 |
| | ODIN | 86.21 | 74.13 | 83.58 | 72.81 | 75.21 | 79.31 | 87.19 | 70.61 | 82 | 74.76 | 82.84 | 74.32 |
| | Energy | 87.55 | 73.91 | 84.38 | 72.58 | 73.46 | 79.83 | 88.53 | 70.17 | 82.54 | 74.69 | 83.29 | 74.24 |
| | GradNorm | 71.08 | 62.5 | 18.99 | 94.06 | 68.35 | 64.57 | 69.62 | 53.13 | 35.56 | 78.99 | 52.72 | 70.65 |
| | Rankfeat | 92.94 | 65.55 | 90.84 | 70.65 | 87.46 | 74.98 | 90.78 | 72.68 | 86.74 | 73.99 | 89.75 | 71.57 |
| | React | 93.15 | 80.88 | 82.3 | 79.63 | 73.02 | 79.88 | 86.07 | 77.9 | 79.01 | 80.54 | 80.83 | 79.77 |
| | GAIA | 15.73 | 97.06 | 33.33 | 94.18 | 63.85 | 89.17 | 16.78 | 97.17 | 15.82 | 97.09 | 29.1 | 94.93 |
| | Our | **15.68** | **97.06** | **33.29** | **94.18** | **63.71** | **89.17** | **16.73** | **97.17** | **15.82** | **97.09** | **29.05** | **94.93** |
| CIFAR100 /WRN40 | MSP | 83.27 | 77.83 | 82.68 | 76.92 | 82.05 | 75.36 | 87.07 | 72.3 | 84.73 | 73.53 | 83.96 | 75.19 |
| | ODIN | 83.44 | 79.85 | 76.68 | 80.32 | 76.91 | 77.84 | 85.81 | 72.5 | 83.42 | 74.95 | 81.25 | 77.09 |
| | Energy | 84.58 | 79.7 | 76.32 | 80.45 | 76.77 | 77.9 | 86.13 | 72.35 | 83.95 | 74.83 | 81.55 | 77.05 |
| | GradNorm | 65.2 | 65.62 | 55.7 | 82.81 | 100 | 4.55 | 98.73 | 14.4 | 77.78 | 44.05 | 79.48 | 42.29 |
| | Rankfeat | 99.97 | 15.4 | 98.79 | 34.34 | 99.04 | 36.01 | 99.71 | 22.18 | 99.47 | 22.49 | 99.4 | 26.08 |
| | React | 94.11 | 67.95 | 87.02 | 67.13 | 88.66 | 65.39 | 89.75 | 64.31 | 89.91 | 63.88 | 89.89 | 65.73 |
| | GAIA | 15.19 | 97.19 | 37.97 | 91.59 | 87.06 | 73.42 | 25.64 | 95.26 | 27.29 | 94.05 | 38.63 | 90.3 |
| | Our | **15.19** | **97.19** | **37.95** | **91.59** | **87.01** | **73.42** | **25.63** | **95.26** | **27.27** | **94.05** | **38.61** | **90.3** |

$$\tau = \sqrt{\sum_{j=1}^{l} \sum_{k=1}^{K} \left( E\left[\epsilon | x^{jk}\right]\right)^2} \qquad (14)$$

where $K$ is the maximum number of channels among all $l$ intermediate layers. We use $E\left[\epsilon | x^{jk}\right]$ in Eq. 11 and Eq. 13 respectively at different levels of label space $Y$. Here $\tau$ represents a confidence score, where a larger $\tau$ indicates a higher likelihood that the input sample is an OOD sample. The pseudocode of our S& I algorithm can be found in **Appendix. A**.

# 4. Experiments

## 4.1. Experimental setup

**Datasets and models:** We followed the experimental setup of GAIA (Chen et al., 2023) and conducted extensive experiments. Specifically, on the CIFAR100 benchmark, we used CIFAR10 as ID datasets (Krizhevsky et al., 2009). We select SVHN (Netzer et al., 2011), TinyImageNet (Liang et al., 2017), LSUN (Yu et al., 2015), Places (Zhou et al., 2017) and Textures (Cimpoi et al., 2014) as OOD datasets. The corresponding backbone models are ResNet34 (He et al., 2016) and WRN40 (Zagoruyko, 2016). On the ImageNet benchmark, we use ImageNet as our ID dataset (Deng et al., 2009). We also selected iNaturalist (Van Horn et al., 2018), SUN (Xiao et al., 2010), Places (Zhou et al., 2017) and Textures (Cimpoi et al., 2014) as OOD datasets. The corresponding backbone model is the pre-trained Google BiT-S (Kolesnikov et al., 2020).

**Baselines and evaluation metrics:** We selected various post-hoc OOD detection methods as our baselines. Among them, MSP (Hendrycks & Gimpel, 2016), ODIN (Liang et al., 2017), Energy-based framework (Liu et al., 2020) are output-based methods. ReAct (Sun et al., 2021) and Rankfeat (Song et al., 2022) are feature representation-based methods. GradNorm (Huang et al., 2021) and GAIA (Chen et al., 2023) are gradient-based methods. Here GAIA is our main competitive baseline. We use FPR95 (false positive rate at 95% true positive rate) and AUROC (area under the receiver operating characteristic curve) as our evaluation metrics (Chen et al., 2023).

## 4.2. Experimental result

**Experiments on CIFAR100 benchmark:** In Tab. 1, we evaluate the OOD detection performance of our S & I algorithm and other baselines on the CIFAR100 benchmark. Since CIFAR100 is a small label space dataset, we use Eq. 11 to obtain the OOD score. Experimental results show that our S & I algorithm achieves the best performance compared with other post-hoc OOD detection methods. Specifically, our method achieves the lowest average FPR95 of 29.05% and 38.61% on ResNet34 and WRN40 models, respectively. For the representative output-based method ODIN, our method achieves 65.15% and 52.47% FPR95 reduction on ResNet34 and WRN40 models, respectively. At the same time, our method achieves 67.63% and 45.24% FPR95 reduction on the ResNet34 model compared with the feature representation-based method Rankfeat and gradient-based method GradNorm, respectively. For the AUROC evaluation metric, our method achieved the highest average AUC of 94.93% on the ResNet34 model. It can be noticed that compared with the main competitive baseline GAIA, our method did not achieve a particularly large improvement on CIFAR100. We believe that this is because the feature distinction between classes in small label space datasets is low, and adversarial attacks may not be able to effectively amplify the difference between ID samples and OOD samples. For the ImageNet-1K dataset with a large label space,

*Table 2.* Experimental result on ImageNet benchmark. Here backbone model is BiT-S. The lower the FPR95, the better the performance, with AUROC behaves inversely. All values are percentages and the best value is bolded.

| Methods | iNaturalist | | Textures | | SUN | | Places | | AVG | |
|---|---|---|---|---|---|---|---|---|---|---|
| | FPR95↓ | AUROC↑ | FPR95↓ | AUROC↑ | FPR95↓ | AUROC↑ | FPR95↓ | AUROC↑ | FPR95↓ | AUROC↑ |
| MSP | 63.93 | 87.57 | 82.66 | 74.45 | 80.24 | 78.22 | 81.43 | 76.71 | 77.06 | 79.24 |
| ODIN | 62.69 | 89.36 | 81.31 | 76.3 | 71.67 | 83.92 | 76.27 | 80.67 | 72.99 | 82.56 |
| Energy | 64.91 | 88.48 | 80.87 | 75.79 | 65.33 | 85.32 | 73.02 | 81.37 | 71.03 | 82.74 |
| GradNorm | 50.03 | 90.33 | 61.42 | 81.07 | 46.48 | 89.03 | 60.86 | 84.82 | 54.7 | 86.3 |
| Rankfeat | 46.54 | 81.49 | **27.88** | 92.18 | 38.26 | 88.34 | **46.06** | **89.33** | 39.69 | 87.84 |
| React | 44.52 | 91.81 | 52.71 | 90.16 | 62.66 | 87.83 | 70.73 | 76.85 | 57.66 | 86.67 |
| GAIA | 29.49 | 93.51 | 40.46 | 92.69 | 34.88 | 92.42 | 48.48 | 88.04 | 38.33 | 91.67 |
| Our | **28.59** | **93.67** | 39.17 | **92.9** | **33.78** | **92.58** | 47.72 | 88.21 | **37.31** | **91.84** |

we can use adversarial attacks to gradually identify OOD samples with high confidence scores, so the improvement is more obvious. We will verify this in the next subsection.

**Experiments on ImageNet benchmark:** In Tab. 2, we evaluate the OOD detection performance of our S & I algorithm and other baselines on the ImageNet benchmark. Since ImageNet is a large label space dataset, we use Eq. 13 to obtain the OOD score. Experimental results show that our S & I algorithm achieves the best performance compared with other post-hoc OOD detection methods. Specifically, our method achieves the lowest average FPR95 of 37.31% on the backbone model BiT-S model. At the same time, our method also achieves the highest AUROC of 91.84%. For the representative output-based method ODIN, our method achieves an FPR95 reduction of 48.88%. At the same time, our method achieves an FPR95 reduction of 6% and 31.79% compared with the feature representation-based method Rankfeat and the gradient-based method GradNorm, respectively. Notably, compared with the main competitive baseline GAIA, our method obtains a 2.66% FPR95 reduction, demonstrating the excellent performance on large label space datasets. Furthermore, to verify the impact of the adversarial attack and the layer-splitting modules (our first two contributions) on OOD detection, we conducted detailed **ablation experiments** in **Appendix. B**. The experimental results demonstrate that both two modules significantly contribute to performance improvement. We also compare the **computational costs** of each method in **Appendix. C**. Finally, **Appendix D** provides additional experiments, including a comparative analysis with SCALE (Xu et al., 2024)—the latest post-hoc OOD detection method proposed in 2024—as well as an investigation into the effects of the adversarial attack learning rate $\eta$.

## 5. Related work

Here we focus on post-hoc OOD detection methods as they can perform OOD detection after the model is deployed without retraining the model or accessing the original train-

ing data. Among them, output-based methods rely on the confidence score of the model output to determine whether the input sample belongs to the training data distribution, which is common in OOD detection based on the maximum softmax probability (MSP) (Hendrycks & Gimpel, 2016). Liang et al. proposed the ODIN algorithm, which utilizes temperature scaling and random perturbations to differentiate the softmax score distributions of ID and OOD samples (Liang et al., 2017). In order to explore the applicability of ODIN in different scenarios, Hus et al. proposed a confidence score decomposition approach and an improved input preprocessing approach based on the existing ODIN algorithm (Hsu et al., 2020). Liu et al. proposed a unified OOD detection framework based on energy scores to replace the traditional softmax score, thereby reducing the effect of overconfident output for softmax scores when inputting OOD samples (Liu et al., 2020). Considering the problem that output-based methods have poor discrimination effect in high-dimensional feature space, feature representation-based methods detect OOD samples by capturing structural information in feature space. Sun et al. proposed the Re-Act (Sun et al., 2021) algorithm based on the analysis of the internal activation pattern of the model to reduce the overconfidence of neural networks on OOD samples. By removing the rank-1 matrix consisting of the largest singular value and its corresponding singular vector in the feature matrix, Song et al. proposed the Rankfeat (Song et al., 2022) algorithm for OOD detection. Gradient-based methods are dedicated to analyzing the gradient information of input samples relative to model parameters (or output of a certain layer) (Huang et al., 2021; Lee & AlRegib, 2020; Igoe et al., 2022). Chen et al. proposed the state-of-the-art GAIA (Chen et al., 2023) algorithm to investigate the different representations of attribution gradients (Simonyan, 2013) on ID and OOD samples for the first time. We further explore the true explanatory pattern representations by layer splitting and adversarial attribution gradient integration to enhance the accuracy of OOD detection.

# 6. Conclusion

In this paper, we contend that non-zero gradient behaviors of OOD samples lack sufficient differentiation, particularly when ID samples are perturbed by random perturbations in high-dimensional spaces, which hampers the accuracy of OOD detection. To tackle this issue, we propose the S & I algorithm. Specifically, we first split the model's intermediate layers and iteratively update adversarial examples layer-by-layer. The attribution gradients of each intermediate layer along the attribution path from adversarial examples to the actual input are integrated to obtain true explanation pattern representations for ID and OOD samples. Experimental results demonstrate that our S & I algorithm achieves superior performance compared to SOTA post-hoc OOD detection methods. The results highlight the effectiveness of S & I algorithm in enhancing the robustness of OOD detection method in dynamic data environments, paving the way for more secure applications in real-world scenarios.

## Impact statement

Ensuring the safe deployment of machine learning models in real-world applications hinges on their ability to recognize unfamiliar or anomalous inputs. This work introduces S & I, a novel Out-of-Distribution (OOD) detection algorithm that integrates adversarial attribution gradients with layer-wise model decomposition. By addressing the limitations of prior gradient-based approaches, S & I captures more faithful explanation pattern representations, achieving state-of-the-art performance on both small- and large-scale benchmarks such as CIFAR100 and ImageNet. Beyond technical innovation, our method contributes to the broader societal goal of trustworthy AI. Robust OOD detection plays a foundational role in safety-critical domains such as autonomous driving, healthcare diagnostics, and real-time decision-making systems, where failing to identify unfamiliar data can lead to harmful or even catastrophic consequences. By improving detection accuracy and stability without requiring retraining, S & I supports more reliable and adaptable AI systems that are better equipped to operate in dynamic, open-world environments.

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

## A. Pseudocode

---

**Algorithm 1** S & I

---

**Input:** Input sample $x$, model $f$ with parameters $\theta$, number of layers $l$, number of iterations $T$, number of channels $K$, image height $R$, image width $S$, loss function $L$, learning rate $\eta$.
**Output:** OOD score $\tau$
 1: $Initialize : x_0^{jk} = x^{jk}$
 2: **for** $i = 1 \rightarrow T$ **do**
 3:     **for** $j = 1 \rightarrow l - 1$ **do**
 4:         Perform adversarial attack by Eq. 9 to get $\triangle x_{i-1}^{jk}$ and $g(x_{i-1}^{jk})$
 5:         Back-propagate adversarial attribution gradients by Eq. 10 or Eq. 12.
 6:         Calculate $E\left[\epsilon|x^{jk}\right]$ by Eq. 11 or Eq. 13 depending on the label space $Y = \{y_m|y_m \in Y\}$.
 7:         Calculate the overall OOD score $\tau$ by Eq. 14.
 8:     **end for**
 9: **end for**
10: **return** OOD score $\tau$

---

## B. Ablation experiments

Here we conduct detailed ablation experiments on the adversarial attack module and layer splitting module of our S& I algorithm. Specifically, we set up three scenarios: a) our model with non-adversarial + layer-splitting, b) our model with adversarial + non-layer-splitting, and c) our model with non-adversarial + non-layer-splitting. The ablation experiment results are as follows:

*Table 3.* S & I (Non-adversarial + Layer Splitting)

| OOD Dataset | FPR95 (%) ↓ | AUROC (%) ↑ |
|---|---|---|
| iNaturalist | 34.75 | 92.48 |
| Textures | 55.05 | 88.73 |
| Sun | 24.85 | 95.38 |
| Places | 40.60 | 91.47 |
| **Average** | **38.81** | **92.02** |

*Table 4.* S & I (Adversarial + Non-layer Splitting)

| OOD Dataset | FPR95 (%) ↓ | AUROC (%) ↑ |
|---|---|---|
| iNaturalist | 28.70 | 93.82 |
| Textures | 61.79 | 85.72 |
| Sun | 32.75 | 92.90 |
| Places | 55.80 | 84.12 |
| **Average** | **44.76** | **89.14** |

*Table 5.* S & I (Non-adversarial + Non-layer Splitting)

| Dataset | FPR95 (%) ↓ | AUROC (%) ↑ |
|---|---|---|
| iNaturalist | 47.05 | 88.92 |
| Textures | 60.11 | 80.76 |
| Sun | 46.35 | 86.62 |
| Places | 67.00 | 79.57 |
| **Average** | **55.13** | **83.97** |

It can be seen that compared with Tab. 3 (Non-adversarial + Layer splitting), Tab. 5 (Non-adversarial + Non-layer splitting) achieved a performance reduction of 16.32% and 8.05% on FPR95 and AUROC, respectively, and performed worse on each dataset, which shows the effectiveness of the layer splitting module. Compared with Tab. 4 (Adversarial + Non-layer splitting), Tab. 5 (Non-adversarial + Non-layer splitting) achieved a performance reduction of 10.37% and 5.17% on FPR95 and AUROC, respectively, and performed worse on other datasets except the FPR95 metric of Textures, which strongly shows the effectiveness of the adversarial attack module. **Therefore, through ablation experiments, we believe that the first two contributions proposed in the introduction are effective.**

## C. Comparison of computational costs

Here we provide a comparison of computational costs between our method and other baselines with the evaluation metric Frame Per Second (FPS). This metric refers to the number of image frames the model can process per second. In OOD detection, FPS reflects the inference speed or real-time capability of the model, serving as a critical metric of algorithm efficiency. A higher FPS indicates faster processing speed.

*Table 6.* FPS comparison of different OOD detection methods (ImageNet)

| Method | FPS |
| --- | --- |
| Our | 4.9019 |
| GAIA | 6.3572 |
| MSP | 16.9090 |
| ODIN | 6.6613 |
| Energy | 16.3934 |
| GradNorm | 8.7750 |
| RankFeat | 4.8416 |
| ReAct | 15.8378 |

From Tab. 6, it can be observed that on the ImageNet dataset, our method is slightly slower than GAIA. However, from the perspective of the main experimental metrics, FPR95 and AUROC, on the ImageNet dataset, our method achieves significant performance improvement. Compared to the computationally similar RankFeat method, our approach not only achieves faster runtime efficiency but also provides more accurate OOD detection. **Therefore, we believe the computational cost of our algorithm is an acceptable trade-off.**

## D. Additional experiments

In this part, we first conduct an ablation study on the hyperparameter learning rate $\eta$. We vary the learning rate in the range 0.0005, 0.001, 0.0015, 0.002. It can be seen that our method consistently achieves high AUROC and low FPR95 scores across all OOD datasets under different settings. The performance remains stable, with only marginal fluctuations observed. This demonstrates that our method is robust to learning rate selection and can generalize well without requiring sensitive hyperparameter tuning. Then, we compare our method with the latest 2024 baseline SCALE (Xu et al., 2024). It can be seen that our method has better performance than SCALE on each OOD dataset. Using CIFAR100 as the benchmark and a learning rate of 0.001, our method shows a 65.32% improvement in FPR95 and a 28.37% improvement in AUROC compared to SCALE.

*Table 7.* Additional experiments

| | Dataset/Model | Methods | SVHN | | LSUN | | TinyImageNet | | Places | | Textures | | AVG | |
| --- | --- | --- | --- | --- | --- | --- | --- | --- | --- | --- | --- | --- | --- | --- |
| | | | FPR95↓ | AUROC↑ | FPR95↓ | AUROC↑ | FPR95↓ | AUROC↑ | FPR95↓ | AUROC↑ | FPR95↓ | AUROC↑ | FPR95↓ | AUROC↑ |
| Ablation experiment on learning rate 0.001 | CIFAR100/ResNet34_0.001 | Our | 15.68 | 97.06 | 33.29 | 94.18 | 63.71 | 89.17 | 16.73 | 97.17 | 15.82 | 97.09 | 29.05 | 94.93 |
| Ablation experiment on learning rate 0.0005 | CIFAR100/ResNet34_0.0005 | Our | 15.67 | 97.05 | 33.32 | 94.17 | 63.72 | 89.15 | 16.71 | 97.17 | 15.74 | 97.1 | 29.03 | 94.93 |
| Ablation experiment on learning rate 0.0015 | CIFAR100/ResNet34_0.0015 | Our | 15.65 | 97.04 | 33.3 | 94.15 | 63.75 | 89.13 | 16.67 | 97.16 | 15.73 | 97.11 | 29.02 | 94.92 |
| Ablation experiment on learning rate 0.002 | CIFAR100/ResNet34_0.002 | Our | 15.69 | 97.03 | 33.4 | 94.13 | 63.74 | 89.13 | 16.72 | 97.16 | 15.7 | 97.12 | 29.05 | 94.91 |
| Comparative experiment with SCALE | CIFAR100/ResNet34 | SCALE | 87.3 | 74.07 | 84.8 | 72.3 | 74.55 | 79.64 | 89 | 70.08 | 83.16 | 73.66 | 83.76 | 73.95 |

