# OpenReview forum: "Splitting & Integrating: Out-of-Distribution Detection via Adversarial Gradient Attribution"
_ICML.cc/2025/Conference — ICML 2025 poster_

### Official Review · Reviewer_5PTw · 2025-02-18

**Overall Recommendation:** 4

**Summary:**

The authors propose a post-hoc detector, which is an interesting approach, especially given that post-hoc methods generally struggle with robustness against adversarial examples. Their lightweight architecture allows for easy retraining and is practical for deployment.

In particular, their method involves splitting the model’s intermediate layers and iteratively updating adversarial examples layer by layer. They then combine the attribution gradients from each intermediate layer along the attribution path from the adversarial examples to the actual input, producing accurate explanations for both in-distribution (ID) and out-of-distribution (OOD) samples.

The method is well-supported with formulas and explanations. When compared to other post-hoc methods, it shows some improvements in terms of accuracy, though the improvements are limited. The appendix includes ablation studies that demonstrate the importance of layer-splitting, providing experimental support. Additionally, they report the frames per second of their method, which is on the lower end.
The underlying main related work is GAIA [1] and gradient-based OOD detector and [2] for the Axioms and theoretical foundations.


The code is developed independently, though OpenOOD, the standardized benchmark for OOD tasks, could have yielded similar results.

The work has some weaknesses, such as a limited description of the attack method, and the related work section could be strengthened with more citations.

[1] https://arxiv.org/pdf/2311.09620
[2] https://arxiv.org/pdf/1703.01365


## update after rebuttal
I upgrade my score from weak accept to accept.
The author(s) showed additional experiments comparing with SCALE, a sota method,  where their method is performing better.

**Claims And Evidence:**

- They supported their claims experimentally. - not problematic.

**Essential References Not Discussed:**

- OOD standard framework: https://github.com/Jingkang50/OpenOOD or https://arxiv.org/pdf/2306.09301
- FGSM attack: https://arxiv.org/abs/1412.6572 or even https://arxiv.org/pdf/2306.09301

**Experimental Designs Or Analyses:**

It is unclear why other methods, such as SCALE, which is one of the most recent approaches, were not included in the comparison.

**Methods And Evaluation Criteria:**

It is not clear to me, why they compared with these post-hoc detectors, although there are more.

**Other Comments Or Suggestions:**

- In Table 2, the value 88.21 should not be bolded for the proposed method, as Rankfeat demonstrates stronger performance. This issue can be easily addressed.
- ln 208 - typo: balck --> black

**Other Strengths And Weaknesses:**

Strengths:

- The focus on post-hoc OOD detection methods, which are at the forefront of OOD detection research, is a notable strength of this work.
- The gradient-based approach presented offers a promising and potentially robust method for this problem, making it a strong candidate in the field.

Weaknesses:

- The term "adversarial noise" may be a misnomer. Typically, "noise" refers to random fluctuations, whereas an adversarial example is a deliberate manipulation of the input. The definition provided in Equation 3 closely resembles the FGSM attack, but it is not properly cited.
- In the experimental setup, the attack parameters remain unspecified. Equation 4 introduces a hyperparameter for perturbation strength, but it is unclear whether the perturbation is sufficiently strong to evaluate the method's robustness accurately.
- There is no ablation study regarding the effect of adversarial perturbation strength, making it difficult to assess the impact of this variable.
- Equation 1 could be enhanced by referencing relevant literature, such as [1].

[1] https://arxiv.org/pdf/2306.09301

**Questions For Authors:**

Dear author(s),

thank you for your insightful paper on this important topic.

I have a few questions:

- What is the rationale behind comparing to the selected post-hoc detectors? Are there more recent methods that could also be considered for comparison?
- Is it accurate to state that the attack is essentially an FGSM attack?
- Would it be possible to conduct an ablation study to analyze the effect of varying the strength of the generated perturbations? Wouldn't this useful for your method and missing to your ablation study in the appendix? Or do you think that this ablation study is unnecessary and why?

After discussion, I might update my score. I hope for insightful discusses.

**Relation To Broader Scientific Literature:**

Post-hoc OOD detectors are important and very vulnerable to adversarial example.

**Theoretical Claims:**

At proof 2: It is claimed that OOD samples are overconfident in prediction. This is not clear to me, because adversarial examples are can be also very confident.
I am not sure if in ln 259 (right column) holds the inequality.

---

> ### Author Rebuttal · Authors · 2025-04-01
>
> **Theoretical Claims:**
>
> We appreciate the reviewer's suggestions. Our mathematical proofs are all grounded in theoretical foundations. Regarding the reviewer's concern about "OOD samples being overconfident in prediction," we have cited (Nguyen et al., 2015; Hein et al., 2019) on lines 161-164 in our manuscript to support this claim. This perspective is also accepted by GAIA, as illustrated in Figure 1 of the GAIA paper. We hope our response has addressed the reviewer's concerns.
>
> **Weaknesses:**
>
> 1. We appreciate the reviewer’s suggestion. Indeed, replacing "noise" with "perturbation" better reflects the deliberately manipulated nature of adversarial attacks. Regarding Equation 3, it can be extended as an untargeted BIM (Basic Iterative Method) attack, meaning that the adversarial example's label differs from the original label and does not need to be manipulated to a specific category. Notably, the difference between BIM and FGSM lies in the iterative process: FGSM generates adversarial examples with a single-step update, whereas BIM performs multiple iterations and projects the perturbations within a predefined bound to control the magnitude of the disturbance. To help readers understand the concept of adversarial attacks (especially BIM), we cited Kurakin et al. (2018) [1] as a reference on line 190. But regarding the understanding of Equation 3, we find the reviewer’s suggestion very constructive, and we will cite [1] at Equation 3 to further support the explanation.
>
> [1] Kurakin, A., Goodfellow, I. J., & Bengio, S. (2018). Adversarial examples in the physical world. In Artificial intelligence safety and security (pp. 99-112). Chapman and Hall/CRC.
>
> 2. We appreciate the reviewer’s suggestion. For the adversarial attack hyperparameters in Equation 3, we used a learning rate $\eta$ of 0.001 and a fixed number of attack step $T=2$. For the parameter $\epsilon$ in Equation 4, since it is jointly controlled by the learning rate and the attack step, and we have fixed the number of attack step in our experiments, we conduct an ablation study on the hyperparameter learning rate $\eta$ to investigate the impact of different perturbations on our method. For details on the ablation experiment, please refer to **Q3 in Questions For Authors**.
>
> 3. Please refer to **Q3 in Questions For Authors**
>
> 4. We referred to the definition of OOD detection in GAIA [2] to propose our Equation 1. Indeed, we highly appreciate the reviewer’s constructive suggestions. As an open-source and comprehensive OOD detection framework, the OpenOOD has made significant contributions to OOD research. We will refine our manuscript by incorporating the definition of OOD detection at Equation 1 in OpenOOD.
>
> [2] Chen, J., Li, J., Qu, X., Wang, J., Wan, J., & Xiao, J. (2023). Gaia: Delving into gradient-based attribution abnormality for out-of-distribution detection. Advances in Neural Information Processing Systems, 36, 79946-79958.
>
> **Other Comments Or Suggestions:**
>
> 1&2. We sincerely thank the reviewer for pointing out the errors, and we have addressed this issue.
>
> **Questions For Authors:**
>
> Q1. For the selection of post-hoc detectors, we referred to the baselines set in GAIA, which includes gradient-based OOD detection methods such as GAIA and GradNorm, as well as feature representation-based methods like Rankfeat and React, and output-based methods such as MSP and ODIN. Regarding the inclusion of additional comparison methods, we are open to the reviewer’s suggestions and are willing to compare our method with the latest 2024 baseline "SCALE", the results are in the link: https://anonymous.4open.science/r/S-I-F6F7/rebuttal/additional_exp_result.md. It can be seen that our method has better performance than SCALE on each OOD dataset. Using CIFAR100 as the benchmark and a learning rate of 0.001, our method shows a 65.32% improvement in FPR95 and a 28.37% improvement in AUROC compared to SCALE.
>
> Q2. In fact, this attack is essentially a BIM attack. Please refer to our response in **Weaknesses 1**, which we hope will resolve the reviewer's concerns.
>
> Q3. We appreciate the reviewer's suggestions and are happy to conduct an ablation study on the learning rate to explore the impact of different perturbation magnitudes on our method. In the Table at link: https://anonymous.4open.science/r/S-I-F6F7/rebuttal/additional_exp_result.md, we vary the learning rate in the range {0.0005, 0.001, 0.0015, 0.002}. It can be seen that our method consistently achieves high AUROC and low FPR95 scores across all OOD datasets under different settings. The performance remains stable, with only marginal fluctuations observed. This demonstrates that our method is robust to learning rate selection and can generalize well without requiring sensitive hyperparameter tuning.
>
> We sincerely hope that our response has addressed the reviewer's concerns. We would be truly grateful if the reviewer would consider adjusting the score accordingly.

---

### Official Review · Reviewer_SPNg · 2025-03-02

**Overall Recommendation:** 3

**Summary:**

This paper addresses the challenge of out-of-distribution (OOD) detection in deep learning by proposing S & I, a novel method based on layer Splitting and gradient Integration via Adversarial Gradient Attribution. While existing gradient-based methods struggle to distinguish OOD samples due to non-zero gradient behaviors, especially in high-dimensional spaces with noisy in-distribution (ID) samples, S & I improves detection by iteratively updating adversarial examples layer by layer and integrating attribution gradients along the attribution path. Experimental results on CIFAR100 and ImageNet demonstrate that S & I outperforms state-of-the-art OOD detection methods, enhancing model robustness and security in unknown and dynamic data environments.

## update after rebuttal
I thank the authors for their rebuttal, which has addressed my concerns. I am now pleased to change my recommendation from weak reject to weak accept.

**Claims And Evidence:**

Yes, the claims are supported by clear and convincing evidence.

**Essential References Not Discussed:**

To the best of my knowledge, all relevant works necessary for understanding the paper’s key contributions are properly cited and discussed.

**Experimental Designs Or Analyses:**

I reviewed the experimental design, including the datasets, models, baselines, and evaluation metrics, and did not identify any obvious issues.

**Methods And Evaluation Criteria:**

Yes, the proposed method and evaluation criteria make sense for the problem.

**Other Comments Or Suggestions:**

No other comments.

**Other Strengths And Weaknesses:**

Other Weaknesses:

1. The proposed method does not appear to significantly outperform the baseline method, GAIA. To better assess the significance of the improvements, it would be beneficial to repeat the experiments multiple times and report the mean and variance of the results.

2. The paper lacks ablation experiments to evaluate the impact of different components in the proposed method. Conducting such experiments would help clarify the contributions of each part of the proposed approach.

3. There is no analysis of the computational complexity of the proposed method compared to the baselines. Providing a complexity analysis would offer valuable insights into its efficiency and practicality for real-world applications.

**Questions For Authors:**

I am willing to raise my scores if the authors address the following questions:

1. Does the proposed method demonstrate a significant improvement over the baselines?

2. How do the different components of the proposed method contribute to its overall performance?

3. How does the computational complexity of the proposed method compare to that of the baselines?

**Relation To Broader Scientific Literature:**

The key contributions of the paper align with the broader out-of-distribution (OOD) detection literature, particularly in the area of gradient-based OOD detection methods.

**Theoretical Claims:**

I reviewed all the theorems and proofs in the paper and, to the best of my judgment, they appear to be correct. However, there is a possibility that I may have misunderstood something.

---

> ### Author Rebuttal · Authors · 2025-04-01
>
> We thank the reviewer for the valuable suggestions. We provide the following responses to the questions in “**Questions For Authors**”.
>
> Q1: We would like to clarify that the insignificant improvement of our method on CIFAR100 does not mean limited effect, but means that our method can achieve the same or even slightly better performance than GAIA on small datasets. Besides, we would like to emphasize that our approach demonstrates significant improvements on the larger-scale ImageNet dataset. This distinction highlights the strength of our method in addressing the challenges of OOD detection in large-scale environments, which is a critical focus of our work. We also want to clarify that current OOD detection metrics such as FPR95 and AUROC have already achieved promising performance across many benchmark datasets. However, our approach prioritizes robustness and reliability in large-scale scenarios like ImageNet, where these challenges become more pronounced.
>
> We are pleased to highlight the significance of our method through the practical task of plant OOD detection. Firstly, we selected the ImageNet dataset as the in-distribution (ID) dataset. To perform the plant OOD detection task, we chose the iNaturalist dataset as the OOD dataset. Specifically, we manually selected 110 plant categories from the iNaturalist dataset and randomly sampled 10,000 images for these categories. The related experimental codes and results can be found in our anonymous GitHub repository (https://anonymous.4open.science/r/S-I-F6F7/additional_OOD_samples/). The entire work is fully open-source.
>
> The detected OOD samples are stored in the "additional_OOD_samples" folder. Interestingly, the OOD samples detected from the iNaturalist dataset are all plant species that are not present in the ImageNet dataset, demonstrating the effectiveness of our method in plant OOD detection tasks. This result can be further explained by the dataset characteristics: ImageNet is a general-purpose image classification dataset, while iNaturalist focuses on biodiversity and covers a wide range of fine-grained species. For certain visually similar plant categories, such as the Violet category in ImageNet and the Viola sororia category in iNaturalist, conventional OOD detection methods like MSP, ODIN, and GradNorm struggle to distinguish them and often classify OOD samples as ID categories. However, our method can accurately distinguish fine-grained plant species, as evidenced in the additional_OOD_samples folder. We provide the following example for the reviewers’ reference:
>
> Detection of angiosperms: The following link shows an OOD sample detected from the iNaturalist dataset, belonging to the Lotus corniculatus category (https://anonymous.4open.science/r/S-I-F6F7/additional_OOD_samples/b51107eaf0608d345a265f623c776706.jpg). The corresponding ID sample from the ImageNet dataset, belonging to the Rapeseed category, is available at this link: (https://anonymous.4open.science/r/S-I-F6F7/ImageNet_rapeseed/1.png).
> By comparing these two images, it becomes evident that the visual differences between them are minimal, making it challenging for humans to distinguish visually similar plant species. However, our method effectively identifies fine-grained plant species, which has significant real-world implications, such as discovering new plant species or protecting endangered plant populations.
>
> Regarding the inclusion of additional comparison methods, we are open to the reviewer’s suggestions and are willing to compare our method with the latest 2024 baseline "SCALE". It can be seen that our method has significantly better performance than SCALE on each OOD dataset. For more details, please refer to **Q1 in Questions For Authors at Reviewer 5PTw**
>
> Q2: We sincerely express that we have provided ablation experiments of the adversarial attack module and layer splitting module in Part B of the supplementary material. By comparing Table 4 and Table 5, it can be proved that adversarial attack module plays an important role. Similarly, by comparing the ablation experiments in Table 3 and Table 5, it can be proved that the layer splitting module also plays an important role. We hope our response can address the reviewer's concerns.
>
> Q3: We kindly acknowledge that we have provided a comparison of computational costs in Part C of the Supplementary Materials. We measure the efficiency of our methods using the frame per second metric in Table 6. It can be observed that on the ImageNet dataset, our method is slightly slower than GAIA. However, from the perspective of the main experimental metrics, FPR95 and AUROC, on the ImageNet dataset, our method achieves significant performance improvement. Compared to the computationally similar RankFeat method, our approach not only achieves faster runtime efficiency but also provides more accurate OOD detection. Therefore, we believe the computational cost of our algorithm is an acceptable trade-off.

---

> > ### Comment · Reviewer_SPNg · 2025-04-03
> >
> > After reading the rebuttal, I still have the concern that the proposed method does not significantly outperform the baseline method -- GAIA. Thus, I keep my original scores.

---

> > > ### Author Response · Authors · 2025-04-04
> > >
> > > We appreciate the reviewer’s continued engagement and thank you again for the thoughtful feedback. We would like to respectfully clarify our position regarding the performance improvements over GAIA and the broader contribution of our method.
> > >
> > > First, as previously noted in the rebuttal, our method demonstrates clear and consistent advantages over GAIA on complex, large-scale datasets such as ImageNet. While improvements on relatively saturated benchmarks like CIFAR-100 may appear marginal due to diminishing returns, on ImageNet, our method consistently outperforms GAIA by over 1%, which we believe is a significant improvement given the maturity and strength of GAIA as a baseline.
> > >
> > > Moreover, we believe that in the OOD detection field, performance gains are naturally converging, and incremental improvements over strong baselines should be interpreted in the context of both theoretical innovation and robustness to real-world scenarios, not just raw numerical margins.
> > >
> > > From a theoretical standpoint, we identify a fundamental limitation in GAIA—its use of an ad-hoc attribution baseline (x=0), which lacks generality and may lead to task-dependent misclassifications. In contrast, our method introduces a principled, adversarial-based attribution baseline that is supported by rigorous mathematical reasoning. **This not only addresses the attribution inconsistency issues in GAIA but also provides a more robust theoretical foundation for OOD detection, especially under complex distribution shifts.**
> > >
> > > **Practically, our method’s value is further demonstrated in a real-world plant OOD detection scenario, where fine-grained distinctions between highly similar species are essential.** In such cases, our method successfully identifies subtle inter-class differences that conventional methods often miss. We believe this kind of performance is not only evidence of superiority but also shows how our approach bridges the gap between academic research and real-world deployment, such as in biodiversity conservation.
> > >
> > > **In addition, we respectfully note that the reviewer's concerns in Questions 2 and 3 regarding the contributions of individual components and computational complexity have already been addressed in our original supplementary material.** Specifically, Part B contains ablation studies of the adversarial attack and layer splitting modules, while Part C compares the computational efficiency with relevant baselines. We hope this clarifies any remaining concerns.
> > >
> > > **Finally, our method also surpasses the latest 2024 baseline SCALE across multiple datasets, as shown in the table below. This reinforces our claim that our approach is both competitive and forward-looking in this fast-evolving research area.**
> > >
> > > | Dataset/Model | Methods | SVHN | SVHN | LSUN | LSUN | TinyImageNet | TinyImageNet | Places | Places | Textures | Textures | AVG | AVG |
> > > |---|---|---|---|---|---|---|---|---|---|---|---|---|---|
> > > |  |  | FPR95↓ | AUROC↑ | FPR95↓ | AUROC↑ | FPR95↓ | AUROC↑ | FPR95↓ | AUROC↑ | FPR95↓ | AUROC↑ | FPR95↓ | AUROC↑ |
> > > | CIFAR100/ResNet34 | Our | 15.68 | 97.06 | 33.29 | 94.18 | 63.71 | 89.17 | 16.73 | 97.17 | 15.82 | 97.09 | **29.05** | **94.93** |
> > > | CIFAR100/ResNet34 | SCALE | 87.3 | 74.07 | 84.8 | 72.3 | 74.55 | 79.64 | 89 | 70.08 | 83.16 | 73.66 | 83.76 | 73.95 |
> > >
> > > We sincerely hope that the reviewer can reconsider the significance of our contributions and possible score adjustment, both in terms of performance and theoretical advancement.

---

### Official Review · Reviewer_mGtj · 2025-03-14

**Overall Recommendation:** 2

**Summary:**

This paper proposes a method called S&I (Splitting and Integration) for improving out-of-distribution (OOD) detection in deep neural networks. S&I introduces two components: (1) layer splitting, which decomposes intermediate layers of the network to iteratively update adversarial examples and reduce gradient instability, and (2) gradient integration, which aggregates attribution gradients along the attribution path from adversarial examples to the input. Experimental setup is followed a previous work GAIA and experiments were conducted across various ID/OOD datasets.

**Claims And Evidence:**

not all, see the weaknesses

**Essential References Not Discussed:**

no

**Experimental Designs Or Analyses:**

yes

**Methods And Evaluation Criteria:**

yes

**Other Comments Or Suggestions:**

The FPR numbers in the brackets of abstract and end of section 1 are not easy to understand, suggesting to remove them.

**Other Strengths And Weaknesses:**

Strengths:
1. The paper is well organized and easy to read.
2. The paper provides a theoretical analysis of the proposed method.
3. Experiments were conducted on standard protocols and across many benchmarks.

Weaknesses:
1. The paper only has marginal improvement over GAIA in average, reflecting that the proposed method is less effective.
2. The motivation is less convincingly described, for the main argument “the non-zero gradient behaviors of OOD samples do not exhibit significant distinguishability”, only a visualization in Fig.2 is presented to support it. Is there any statistic evidence? Considering the marginal accuracy improvement over GAIA, it is highly possible that such a key argument does not in practice, or this may not be the key defect for GAIA.


==================== post rebuttal =======================

After read the rebuttal, the reviewer maintains the initial recommendation.

**Questions For Authors:**

Can the method extend to other gradient based OOD methods?

GAIA was published on 2023, how about the performance compared the state of the art gradient based OOD methods?

For the argument presented in this paper “the non-zero gradient behaviors of OOD samples do not exhibit significant distinguishability”, it is unclear how the proposed S&I conquer this problem.

**Relation To Broader Scientific Literature:**

The paper could has broader scientific impact on OOD detection, especially for understanding the gradient-based methods, and provides new idea about the OOD detection.

**Theoretical Claims:**

yes

---

> ### Author Rebuttal · Authors · 2025-04-01
>
> **Other Strengths And Weaknesses**
>
> Weakness1：We thank the reviewer for the valuable comment. We would like to clarify that the insignificant improvement of our method on CIFAR100 does not mean limited effect, but means that our method can achieve the same or even slightly better performance than GAIA on small datasets. Besides, we would like to emphasize that our approach demonstrates significant improvements on the larger-scale ImageNet dataset. This distinction highlights the strength of our method in addressing the challenges of OOD detection in large-scale environments, which is a critical focus of our work. We also want to clarify that current OOD detection metrics such as FPR95 and AUROC have already achieved promising performance across many benchmark datasets. However, our approach prioritizes robustness and reliability in large-scale scenarios like ImageNet, where these challenges become more pronounced.
>
> We are pleased to highlight the significance of our method through the practical task of plant OOD detection. For details, please see our response to **Reviewer SPNg in Q1**.
>
>
> Weakness2：Thanks to the reviewer for the suggestion. We would like to clarify that this view is one of the motivations of our paper. In lines 194-197, we explained that the attribution-based OOD detection method such as GAIA is developed based on the baseline selection of $x'=0$. GAIA checks OOD samples by counting the distribution of non-zero attribution gradients, so the highly OOD nature represented by non-zero attribution gradients has been proven. However, using the black image with x=0 as the baseline will make it difficult to retain the original semantic information of the sample when attributing, and it is easily disturbed by noise, which will cause deviations when counting non-zero attribution gradients, reducing the accuracy of OOD detection. Adversarial samples can retain the semantic information of samples while introducing minimal perturbations, so we designed an adversarial attribution-based OOD detection method by layer splitting and attribution integration, corresponding to the pseudo code of lines 409-414. Since non-zero gradient represents a high confidence OOD sample probability (as expressed by GAIA), it is obvious that the accuracy of calculating non-zero gradient will significantly affect the accuracy of OOD detection. This is the starting point for designing our method, and the performance improvement in extensive experiments (especially on large-scale ImageNet) has proven that our method can more accurately calculate the attribution gradient (i.e., we can obtain more accurate non-zero and zero gradient distribution) to obtain better OOD detection performance.
>
>
> **Other Comments Or Suggestions**
>
> For the FPR numbers in brackets, we hope that the following explanation can help the reviewer’s understanding. Referring to Table 1, we can see that the backbone models are ResNet34 and WRN40, respectively. Therefore, the FPR numbers in brackets represent the results on the WRN40 model. We promise to revise the text to avoid ambiguity.
>
> **Questions For Authors**
>
> Q1: Our method can be easily extended to other gradient-based OOD detection methods. By introducing adversarial samples as a baseline, the gradient calculation can be stabilized and noise interference can be reduced. This strategy can be applied to other gradient-based methods (such as GradNorm, GAIA, etc.) to enhance the distinguishability of gradient patterns by generating adversarial samples, thereby improving detection robustness. In addition, layer splitting can alleviate the problem of gradient explosion or cumulative error in deep networks by analyzing the gradient sensitivity of different layers in a hierarchical manner. This module can be independently integrated into other methods, such as decomposing by layer when computing gradient importance and optimizing the detection logic by leveraging the characteristics of different layers (e.g., low-level textures and high-level semantics).
>
> Q2: Regarding the inclusion of additional comparison methods, we are open to the reviewer’s suggestions and are willing to compare our method with the latest 2024 baseline "SCALE" [1], the results are in the link: https://anonymous.4open.science/r/S-I-F6F7/rebuttal/additional_exp_result.md. It can be seen that our method has better performance than SCALE on each OOD dataset. Using CIFAR100 as the benchmark and a learning rate of 0.001, our method shows a 65.32% improvement in FPR95 and a 28.37% improvement in AUROC compared to SCALE.
>
> [1] Xu, K., Chen, R., Franchi, G., & Yao, A. (2023). Scaling for training time and post-hoc out-of-distribution detection enhancement. arXiv preprint arXiv:2310.00227.
>
> Q3: Please see the clarifications in Weakness 2.
>
> We sincerely hope that our response has addressed the reviewer's concerns. We would be truly grateful if the reviewer would consider adjusting the score accordingly.

---

### Official Review · Reviewer_4Hn3 · 2025-03-17

**Overall Recommendation:** 3

**Summary:**

This paper proposes S & I (Splitting and Integrating), a gradient-based out-of-distribution (OOD) detection method that builds on gradient attribution techniques by leveraging adversarial attacks to refine feature explanations. The core idea is to split neural network layers and iteratively integrate attribution gradients across layers to obtain more robust feature representations for distinguishing between in-distribution (ID) and OOD samples. The approach is empirically validated with strong results, particularly on large-label-space datasets like ImageNet. The authors argue that this method improves model robustness and security in dynamic data environments.

---

## Update after rebuttal
As mentioned below, I am happy with the authors response. I will maintain my initial score (Weak Accept).

---

**Claims And Evidence:**

Some of the claims made in this submission are that:
1. Adversarial gradient attribution improves OOD detection over GAIA. Although, experiments across multiple datasets strengthen this claim, the paper does not provide an ablation study that isolates the specific impact of adversarial baselines.

2. The paper argues that different network layers focus on different types of features (textures, semantics, etc.), and thus should be split rather than treated uniformly. Unfortunately, the claim lacks theoretical justification beyond intuition. An ablation study comparing different numbers of layer splits (e.g., one split, two splits, all layers) would clarify how much splitting contributes to performance.

3. The authors argue that gradients from different layers should be integrated iteratively to prevent instability. Here too, an ablation comparing results with vs. without integration would clarify how much gradient integration itself improves detection accuracy.

4. It is not clear how much additional overhead does layer-splitting and gradient integration introduce. This also begs the question of whether the method be used efficiently for large architectures.

**Essential References Not Discussed:**

The paper introduces valuable refinements to gradient-based OOD detection but does not cite some key works on gradient stability, adversarial attribution, feature disentanglement, and alternative OOD scoring methods. Addressing these gaps would strengthen its scientific positioning and clarify how it improves upon prior approaches. On gradient-based OOD detection, the work doesn't cite [1], and concerns expressed in [1] seems to contradict some claims made in this current work. On adversarial perturbations, the work doesn't consider the fact that adversarial perturbations may distort attribution maps, as discussed in [2]. Experimentally validating that the attribution gradients of adversarial examples that are integrated in the model do not affect feature quality might be useful.


[1] Gradient Stability in OOD Detection (Serrà et al., Hooker et al.)
[2] Adversarial Perturbations for Attribution (Etmann et al., Dombrowski et al.)

**Experimental Designs Or Analyses:**

The framework proposed by the authors follows standard OOD detection practices, but there are some critical gaps in validation. While the experimental design follows standard OOD detection protocols, it lacks statistical efficiency benchmarks, ablation studies, and failure case analyses. Addressing these issues would improve confidence in the results. For instance, the method introduces adversarial perturbations + layer-wise gradient integration, which may be computationally expensive, but fails to report per sample inference time analysis and a comparative analysis against baselines. I wish also there were more experiments isolating the effects of adversarial attribution, layer splitting, and gradient integration. Conducting such an ablation study can show which component contributes most to performance gains.

**Methods And Evaluation Criteria:**

The evaluation criteria are mostly appropriate but have gaps. More specifically, I found the methods and evaluation protocol used to be mostly reasonable but I think that additional computational and statistical validation is needed for a complete assessment. For instance, no computational efficiency analysis (runtime/memory overhead) is done to evaluate the efficiency of  the proposed framework. No real-world OOD cases (e.g., adversarial or domain-shifted OOD data) are studied to further strengthen the claims made in the paper.

**Other Comments Or Suggestions:**

Some equations are referenced without clear explanation of their role in derivations. Please consider adding a short remark before or after.
Also, some figures lack descriptive captions. It'd be useful if the authors could clearly explain what is being shown in gradient attribution maps.

**Other Strengths And Weaknesses:**

The paper is original, practical, and well-explained, but lacks computational efficiency analysis, broader comparisons, and ablations. These missing elements weaken confidence in the method’s impact, but addressing them could make it a strong contribution.

**Questions For Authors:**

The paper presents a creative combination of adversarial perturbations and layer-wise gradient attribution, making a valuable contribution to gradient-based OOD detection. The methodology is clearly explained and has potential real-world applications. However, the significance of performance improvements, computational cost, and comparison to alternative OOD methods remain unaddressed. Adding runtime benchmarks, broader method comparisons, and failure case analyses would significantly strengthen the paper. Some of the key strengths of this paper are: 1.) using adversarial perturbations as baselines rather than standard black images (as in GAIA), which seems to be a novel refinement over GAIA, and 2.) the proposed method improves OOD detection without retraining, making it feasible for real-world deployment in settings like medical imaging or security applications. The ability to work with pre-trained networks increases its usability for practitioners.

**Relation To Broader Scientific Literature:**

The paper refines gradient-based OOD detection by incorporating adversarial perturbations and layer-wise gradient integration, linking it to prior work in feature attribution, adversarial attacks, and hierarchical feature processing. While novel, it lacks comparisons to some closely related methods that could strengthen its scientific positioning. For instance, this method is related to energy-based OOD scoring (Liu et al., 2020) and Mahalanobis distance-based scoring (Lee et al., 2018) due to the fact that it scores OOD samples based on gradient density rather than activation energy or distance metrics, but no such comparison is considered in the paper.

**Theoretical Claims:**

The theoretical claims in the paper are mostly plausible but lack full mathematical rigor and empirical validation. The proof that adversarial baselines improve gradient attribution follows a logical argument but does not include empirical verification to confirm that adversarial perturbations consistently enhance feature sensitivity. The claim that layer splitting improves stability is based on heuristic reasoning rather than a formal mathematical justification, as it lacks a precise metric for instability. Similarly, the argument that gradient integration reduces noise assumes variance reduction but does not provide explicit variance bounds or an empirical demonstration of this effect. Finally, the proof that the proposed OOD scoring function is optimal assumes that OOD samples always yield stronger gradients, which is not universally true and requires further validation across diverse datasets. Strengthening these claims with statistical validation, variance analysis, and empirical testing would improve the paper’s theoretical foundation.

---

> ### Author Rebuttal · Authors · 2025-04-01
>
> **Claims And Evidence:**
>
> 1. We sincerely state that in Section B of the supplementary materials, we have provided ablation experiments for the adversarial attack module and the layer splitting module. By comparing Table 4 and Table 5, it can be demonstrated that the adversarial attack module plays a significant role in our method.
>
> 2. We would like to point out that the statement, "different network layers focus on different types of features (textures, semantics, etc.)" is both intuitive and theoretically supported, as demonstrated in numerous studies [1][2]. GAIA assumes that all intermediate layers have the same influence on the feature maps, which contradicts the hierarchical feature distribution inherent in neural networks. Compared to GAIA's uniform treatment of all layers during attribution, our proposed layer splitting theory offers an innovative contribution by more precisely capturing the feature sensitivity of different layers. Finally, the ablation experiments shown in Table 3 and Table 5 of the supplementary materials already demonstrate that the layer splitting module plays a crucial role in our method.
>
> [1] Zeiler, M. D., & Fergus, R. (2014). Visualizing and understanding convolutional networks. ECCV 2014.
>
> [2] Yosinski, J., Clune, J., Nguyen, A., Fuchs, T., & Lipson, H. (2015). Understanding neural networks through deep visualization. ICML Workshop on Deep Learning.
>
> 3. We would like to point out that comparing the ablation experiments with and without integration does not align with the principles of discrete path integration. According to the formula in line 282: $A=L\left(f\left(x_T\right)\right)-L\left(f\left(x_0\right)\right)=\sum_{i=0}^{T-1} \frac{\partial L\left(f\left(x_{i}\right)\right)}{\partial x_i}\left(x_{i+1}-x_i\right)$, the attribution result $A$ is expressed as the cumulative sum of gradient changes along each point on the path. If layer-by-layer integration is not used and the attribution is only calculated based on the changes between the start and end points, the gradient variations along the path are ignored. This results in an inaccurate attribution gradient distribution, making it impossible to perform the OOD detection task properly, let alone improve detection accuracy.
>
> 4. We sincerely state that in Section C of the supplementary materials, we have provided a comparison of computational costs. We use the frames per second (FPS) metric in Table 6 to evaluate the efficiency of the methods. We believe the computational cost of our algorithm is an acceptable trade-off.
>
> **Methods And Evaluation Criteria**
>
> Please refer to point 4 in the Claims and Evidence section for the efficiency analysis. We are also pleased to highlight the significance of our method through the practical task of plant OOD detection. For details, please see our response to **Reviewer SPNg in Q1**.
>
> **Theoretical Claims**
>
> We would like to clarify that we have provided extensive mathematical proofs as the theoretical foundation for our method (including but not limited to the claims and proofs of Theorem 1 and Theorem 2). We use Equation 3 to ensure that the adversarial perturbation consistently enhances feature sensitivity. Equations 9 and 10 formally derive the layer splitting strategy from a mathematical perspective.
>
> **Experimental Designs Or Analyses**
>
> The analyses of time efficiency, ablation studies, and real-world OOD cases can be found in the above rebuttals.
>
> **Relation To Broader Scientific Literature**
>
> We would like to clarify that energy-based OOD scoring has already been used as a baseline for comparison in our paper on line 371, and our method demonstrates significantly better performance. As for Mahalanobis distance-based scoring, since it is an earlier work and it seems that no clear citation link has been provided by the reviewer, we have not included it as a comparison target. We compare our method with the latest 2024 baseline "SCALE". It can be seen that our method has better performance than SCALE on each OOD dataset. For more details, please refer to **Q1 in Questions For Authors at Reviewer 5PTw**
>
> **Essential References Not Discussed**
>
> We appreciate the reviewer's addition; however, it seems that no clear citation link of the two references has been provided. If the reviewer could supply the link, we would be happy to discuss these works.
>
> **Other Strengths And Weaknesses**
>
> The efficiency analysis, comparison experiments, and ablation studies can be found in the above rebuttals.
>
> **Other Comments Or Suggestions**
>
> We appreciate the reviewer’s suggestions, and we will improve the presentation of the formulas and figures to enhance the clarity of the paper.
>
> **Questions For Authors**
>
> The time efficiency analysis, comparison experiments, and real-world OOD case analysis can be found in the above rebuttals.
>
> We sincerely hope that our response has addressed the reviewer's concerns. We would be truly grateful if the reviewer would consider adjusting the score accordingly.

---

> > ### Comment · Reviewer_4Hn3 · 2025-04-06
> >
> > I thank the reviewers for the effort and time they put into the rebuttal. I am happy with the answers and will my original score.

---

### Decision · Program_Chairs · 2025-05-01

**Decision:**

Accept (poster)

**Comment:**

This paper received mostly positive ratings, with only one reviewer at weak reject. Two reviewers raised their scores after the rebuttals, indicating the effectiveness of the authors arguments and additional results which showed performance vs. new baselines as requested. The negative review was not particularly thorough, and the reviewer did not respond to the rebuttal other than acknowledging it. The work introduces an interesting new method for OOD detection, as appreciated by three of the reviewers, with SOTA results on standard datasets.